# Analysis of Codon Usage Bias in Xyloglucan Endotransglycosylase (XET) Genes

**DOI:** 10.3390/ijms24076108

**Published:** 2023-03-24

**Authors:** Bo Xiong, Tie Wang, Shengjia Huang, Ling Liao, Xun Wang, Honghong Deng, Mingfei Zhang, Jiaxian He, Guochao Sun, Siya He, Zhihui Wang

**Affiliations:** 1College of Horticulture, Sichuan Agricultural University, Chengdu 611130, China; 2Institute of Pomology and Olericulture, Sichuan Agricultural University, Chengdu 611130, China

**Keywords:** xyloglucan endotransglycosylase, codon usage bias, optimal codons

## Abstract

Xyloglucan endotransglycosylase (*XET*) genes are widely distributed in most plants, but the codon usage bias of *XET* genes has remained uncharacterized. Thus, we analyzed the codon usage bias using 4500 codons of 20 *XET* genes to elucidate the genetic and evolutionary patterns. Phylogenetic and hierarchical cluster analyses revealed that the 20 *XET* genes belonged to two groups. The closer the genetic distance, the more similar the codon usage preference. The codon usage bias of most *XET* genes was weak, but there was also some codon usage bias. AGA, AGG, AUC, and GUG were the top four codons (RSCU > 1.5) in the 20 *XET* genes. *CitXET* had a stronger codon usage bias, and there were eight optimal codons of *CitXET* (i.e., AGA, AUU, UCU, CUU, CCA, GCU, GUU, and AAA). The RSCU values underwent a correspondence analysis. The two main factors affecting codon usage bias (i.e., Axes 1 and 2) accounted for 54.8% and 17.6% of the total variation, respectively. Multiple correspondence analysis revealed that *XET* genes were widely distributed, with Group 1 genes being closer to Axis 1 than Group 2 genes, which were closer to Axis 2. Codons with A/U at the third codon position were distributed closer to Axis 1 than codons with G/C at the third codon position. *PgXET*, *ZmXET*, *VlXET*, *VrXET*, and *PcXET* were biased toward codons ending with G/C. In contrast, *CitXET*, *DpXET*, and *BrpXET* were strongly biased toward codons ending with A/U, indicating that these *XET* genes have a strong codon usage bias. Translational selection and base composition (especially A and U at the third codon position), followed by mutation pressure and natural selection, may be the most important factors affecting codon usage of 20 *XET* genes. These results may be useful in clarifying the codon usage bias of *XET* genes and the relevant evolutionary characteristics.

## 1. Introduction

Codon usage bias, which is usually the result of adaptive evolution, refers to the unequal frequency of codons during translation [1]. Codon usage bias is of great importance in the chemical distances between amino acids, as the occurrence of the errors also relies on the frequency of different codons [2]. Even within a genome, different codon usage preferences are observed among genes [3]. Recent studies have revealed that specific synonymous codon usage biases affect protein folding [4,5]. When the gene expression level was higher, the evolutionary rates and selective pressure were lower, but the codon usage bias was strong [6]. Except for tryptophan (Trp) and methionine (Met), all amino acids are encoded by at least two synonymous codons [7]. However, the synonymous codons do not occur with equal frequency, and a specific codon usage pattern is referred to as codon usage bias [8]. Mutation, natural selection, and random drift are the three major factors contributing to codon usage bias [9,10]. There are some additional factors, including gene length and expression level, tRNA abundance, RNA stability, protein structure, and GC content [3,7,11]. Codon usage bias is believed to be a result of a balance between translation selection and mutation pressure [12]. Molecular evolution studies have shown that codon usage bias exists in a wide range of species, from prokaryotes to eukaryotes, and may have profound implications for genome evolution [2].

The synonymous codon selection of amino acids in different plant genomes is not random. There is a synonym codon usage (SCU) bias. Previous research has shown that SCU bias is related to genome size [13], gene length [14], gene translation initiation signal [15], and GC compositions [16]. Therefore, the investigation of codon bias patterns is important in gene biology studies.

Xyloglucan endotransglucosylase/hydrolase (XTH) is a key xyloglucan modifying enzyme belonging to the glycoside hydrolase 16 (GH16) family, which catalyzes the cleavage and polymerization of xyloglucan molecules to modify the celluline–xyloglucan complex structure of seed germination [17,18], and cell walls [19] and elongation [20,21]. XTHs, which are directly involved in the initial assembly and subsequent restructuring of primary cell walls [22,23], exhibit two diverse catalytic activities: xyloglucan endohydrolase (XEH) and xyloglucan endotransglycosylase (XET) [24]. XTH proteins are essential for plant growth and differentiation [25]. Unique members of the *XTH* gene family are involved in many abiotic stress responses [26].

XET (EC 2.4.1.207) was found in all land plants [27]. Previous studies have found that the *XET* expression levels are correlated with cell expansion [28,29], and cell growth is promoted by XTHs [30,31]. XETs release a smaller xyloglucan from the reducing end of a donor xyloglucan, then another xyloglucan chain is added to the newly generated free end [2]. XET is one of the most important parts of ancient machinery regulating cell wall modifications [32]. In different species, the XET gene is involved in seed germination [18], cell wall construction [33], and abiotic stress response [21,26], indicating that the function of the *XET* gene is significantly different.

In recent years, many *XET* genes have been cloned. There is limited information on the codon usage bias of *XET* genes. Here, we analyzed the codon usage bias of 20 different *XET* genes and explored factors that may be related to this codon usage bias. A comprehensive analysis of *XET* gene codon usage bias will be important for understanding its biological significance and providing theoretical advice for optimizing the codons of *XET* genes.

## 2. Results

### 2.1. Clustering Analysis

The coding sequences of 20 different *XET* genes were analyzed using the neighbor-joining method to construct a phylogenetic tree. The following genes were observed to belong to Group 1: *Populus euphratica XET* (*PeXET*), *Paeonia ostii* var. lishizhenii *XET* (*PoXET*), *Dahlia pinnata XET* (*DpXET*), *Citrus* cultivar Huangguogan *XET* (*CitXET*), *Medicago truncatula XET* (*MtXET*), *Actinidia deliciosa XET*-5 (*AdXET*-5), *Sinopodophyllum hexandrum XET* (*ShXET*), *Camellia sinensis XET* (*CsXET*), *Brassica rapa* subsp. pekinensis *XET* (*BrpXET*), *Brassica rapa* (field mustard) *XET* (*BrfXET*), *Pyrus pyrifolia XET* (*PpXET*), *Rosa chinensis XET* (*RcXET*), and *Populus alba* × *Populus tremula* var *XET* (*Pa* × *Pt XET*) (Figure 1). The ranges in the effective number of codons (ENc) values and GC3 contents were relatively narrow (i.e., 47.0–60.7 and 27.4–61.2, respectively) (Table 1). These results suggest that the similarity in codon usage increases with increasing evolutionary relatedness of genes. Group 2 consisted of seven *XET* genes, including *Pyrus communis XET* (*PcXET*), *Gossypium barbadense XET* (*GbXET*), *Zea mays XET* (*ZmXET*), *Actinidia deliciosa XET* (*AdXET*), *Vigna radiata* cultivar T44 *XET* (*VrXET*), *Vigna luteola XET* (*VlXET*), and *Pennisetum glaucum XET* (*PgXET*) (Figure 1). In contrast to the corresponding Group 1 data, the ranges of the ENc values and GC3 contents were relatively wide (i.e., 31.1–56.5 and 43.2–97.9, respectively) (Table 1). These observations indicated that the codon usage was different among these evolutionarily related genes.

The relative synonymous codon usage (RSCU) values of the 20 *XET* genes underwent hierarchical cluster analysis (Figure 2). The *VlXET*, *VrXET*, *PgXET*, *ZmXET*, and *PcXET* genes always clustered together and had high RSCU values for the following codons: CUG, GGC, CGG, CCG, AAC, ACG, UUC, CAG, AUC, CAC, GCC, AAG, ACC, UCC, GUC, GGG, UAC, GUG, CUC, GAC, GAG, CCC, AGC, and CGC. Another cluster comprised *CitXET*, *MtXET*, *DpXET*, *BrpXET*, and *BrfXET*. *CitXET* and *DpXET* genes were relatively closely related (Figure 1) and exhibited similar codon usage (Figure 2). These results suggested that different *XET* genes had diverse codon usage patterns, but there was also some codon usage bias.

### 2.2. Analysis of Codon Usage Bias in 20 XET Genes

The average GC content of the 20 analyzed *XET* genes was 50.9% (range: 38.8–69.5%). The GC content was highest in the third codon position (i.e., 58.2%). Additionally, the mean ENc value was 48.60 (range: 31.14–60.71). With ENc values <35, a strong codon bias was associated with *PgXET*, *VlXET*, *VrXET*, and *ZmXET*. These data suggested that some *XET* genes exhibited a directional codon usage, reflecting the codon usage bias of a few *XET* genes (Table 1). 

The top four codons (RSCU > 1.5) of the 20 *XET* genes were AGA and AGG (encoding arginine), AUC (encoding isoleucine), and GUG (encoding valine) (Table 2). Some codons were associated with high usage bias (RSCU > 1.0), including AGA, AGG, AUC, GUG, UAC, UCC, GCU, UUC, CUC, GCC, AAC, GAG, CUG, AAG, UCU, ACU, CCA, CAG, CAC, ACC, GGA, GAC, UUG, UGC, GGC, CUU, CCC, and ACA. UAU, AUA, CGU, CUA, and CGA (RSCU < 1.0) were the five least frequent codons. For the 20 *XET* genes, more than 60% of AGA codons had RSCU greater than 1.5, and the AGA codon of *CitXET* had the highest RSCU value (3.90) (Table 2 and Table 3). There were 21 codons that had ΔRSCU values > 0.08, which were defined as preferred codons in the 20 *XET* genes, including UUU, UUA, UUG, CUU, AUU, AUA, GUU, GUA, UCU, AGU, CCU, CCA, ACU, ACA, GCU, GCA, GCG, UAU, CAU, CAA, AAU, AAA, GAU, GAA, UGU, CGU, CGA, AGA, AGG, GGU, and GGA (Table 2). Additionally, AGA, GCU, CUU, UCU, CCA, UCA, GUU, UAC, AAA, and AUU were the most common codons in *CitXET* (RSCU > 1.5) (Table 3). Based on a comparison and screening, eight optimal codons of *CitXET* were identified (i.e., AGA, AUU, UCU, CUU, CCA, GCU, GUU, and AAA). Thus, *CitXET* is biased toward synonymous codons with A/U at the third codon position.

The RSCU values underwent a correspondence analysis. The two main factors influencing codon usage bias were represented by Axis 1 and Axis 2 (Figure 3) and contributed 54.8% and 17.6% of the total variation, respectively. A principal component analysis was used to examine the relationship between amino acid composition and codon usage bias. There was a distinct positive correlation between Axis 1 and C3s (*r* = 0.966, *p* < 0.01), G3s (*r* = 0.866, *p* < 0.01), and GC3s (*r* = 0.963, *p* < 0.01). In contrast, Axis 1 was negatively correlated with A3s (*r* = −0.935, *p* < 0.01) and T3s (*r* = −0.861, *p* < 0.01). There was also a negative correlation between GC3s and ENc (*r* = −0.837, *p* < 0.01). All correlation coefficients were greater than 0.83 (Figure 4). The multiple correspondence analysis revealed that the *XET* genes were widely distributed, with Group 1 genes closer to Axis 1 than the Group 2 genes, which were closer to Axis 2 (Figure 5). The codons with A/U at the third codon position were distributed closer to Axis 1 than the codons with G/C at the third codon position (Figure 6). These results suggested that the base composition, especially for codons with A/U at the third codon position, had some effect on codon usage bias. Thus, many factors might influence codon usage bias. Axis 1 represented the main factor affecting codon usage bias, but the base composition was another very important factor.

### 2.3. Base Composition Affects the Formation of Codon Usage Bias

To clarify the relationships among the two groups of *XET* genes regarding the third codon positions of synonymous codons, the RSCU value for each codon underwent a multidimensional preference analysis. The Group 1 and Group 2 genes were closely associated with codons ending with A/U and G/C, respectively. The relatively short distance between *PgXET*, *ZmXET*, *VlXET*, *VrXET*, and *PcXET* and GC3s suggested that the codon usage might be biased toward G/C. However, *CitXET*, *DpXET*, and *BrpXET* were very close to AU3s, suggesting a strong codon usage bias toward codons ending with A/U (Figure 5 and Figure 6). Correlation analysis between CAI and ENc was used to determine the effects of translation selection and mutation pressure on codon usage bias of *XET* genes. The results showed that CAI was significantly negatively correlated with ENc (*r* = −0.737, *p <* 0.01) (Figure 7), reflecting that translational selection contributed more to codon usage patterns of the 20 *XET* genes than mutations.

## 3. Discussion

The biological process of genetic information from mRNA to protein depends on codon coding [34]. Codon is an important part of the output of nucleic acid information [35,36]. Previous research has found that codon usage bias is mainly influenced by mutation pressure and natural selection [37]. The codon usage bias exists widely in different species [38,39]. However, there are significant differences among different species in the main factors that guide densification code usage bias.

The GC composition has been shown to drive amino acids and codon usage that are closely related to the usage pattern of the third codon base (GC3) [40]. There was a wide range of the GC3s values of GC content in *XET* codons (27.4–97.9%). The GC3 content of *PgXET*, *VlXET*, *VrXET*, and *ZmXET* was more than 93%, indicating that these *XET* genes preferred significantly to end with G/C base, while *CitXET*, *BrfXET*, *BrpXET*, and *MtXET* preferred the codons ending with A/T(U) base. These results suggested that mutation pressure was the main factor affecting codon usage of 20 *XET* genes.

In this study, phylogenetic and hierarchical cluster analyses produced similar results. Compared to the corresponding Group 1 data, the ranges of ENc values and GC3 contents were relatively wide (i.e., 31.1–56.5 and 43.2–97.9, respectively) (Table 1). These observations indicated that codon usage was different, and different *XET* genes had different codon usage patterns, but there was also some codon usage bias.

ENc and CAI are two parameters related to gene expression level. A subsequent series of analyses confirmed that most *XET* genes have an ENc value > 35, suggesting these genes exhibit a general codon usage pattern. The exceptions were *ZmXET*, *VlXET*, *VrXET*, and *PgXET.* The ENc is one of the most important factors affecting codon usage preferences. An ENc value approaching 20 indicates a strong codon usage bias. In contrast, an ENc value close to 61 implies that the codons are used relatively equally [3]. Many factors contribute to biased synonymous codon usage in the 20 *XET* genes. For example, base composition influences codon usage bias (e.g., *CitXET*, *DpXET*, and *BrpXET*), especially regarding codons with A/U at the third codon position.

In order to evaluate the synonymous codon usage pattern with amino acids of different *XET* genes, the codon usage bias of various codons was detected by RSCU, which reflected the frequency of a specific codon relative to that of the synonymous codons. A relatively strong codon usage bias is indicated by an RSCU value > 1. In contrast, an RSCU value < 1 indicates that a particular codon is used less frequently than the other synonymous codons [41]. In this study, AGA, AGG, AUC, and GUG were the top four codons (RSCU > 1.5) in the 20 *XET* genes. A comparison of the RSCU values of the other 19 *XET* genes revealed a similar codon usage bias, while *CitXET* had a stronger codon usage bias. There were eight optimal codons of *CitXET* (i.e., AGA, AUU, UCU, CUU, CCA, GCU, GUU, and AAA), suggesting *CitXET* was biased toward the synonymous codons with A or U at the third codon position. We found that XET activity was specific to the elongation of root cells during citrus seedling etiolation in previous research [42]. Consequently, the codon usage bias may be very closely related to the functionality of *CitXET*. In order to confirm this assumption, further research is needed.

The correlation between the first two spindles (Axis 1 and Axis 2) and the nucleotide content of 20 XET genes was analyzed. The results showed that there were various significant correlations between the two spindle axes and nucleotide content. In addition, there was a significant negative correlation between Gravy values with Axis 1 and a significant positive correlation between Gravy with Axis 2. In contrast, there was a positive correlation between Aromo and Axis 1 and a significant negative correlation between Aromo and Axis 2 (Figure 4). These results suggested that Axis 1 and Axis 2 had important roles in shaping *EXT* codon usage patterns.

The CAI, with values ranging from 0 to 1, is another effective measure of codon usage bias. The higher the CAI value, the better the adaptability of the sequence [6]. CAI value approaching 1 indicates a strong codon bias [3]. In the present study, with the exception of *PgXET* and *ShXET*, the analyzed genes had CAI values between 0.2 and 0.3. These results indicated that the expression levels of the *XET* genes were not randomly variable. In previous studies, the correlation between CAI and ENc values was used to demonstrate the effects of translational selection and mutation pressure on codon usage bias [43,44,45]. A correlation coefficient approaching −1 suggests that the translation selection effect is greater than that of mutations on codon usage bias [7]. Our correlation analysis showed that CAI was significantly negatively correlated with ENc (*r* = −0.737, *p <* 0.01) (Figure 7), reflecting that translational selection contributed more to codon usage patterns of the 20 *XET* genes than mutations. Similar results were reported by Qiu et al. [46], but these observations were inconsistent with the findings of other studies, which suggested that mutations had a stronger effect on codon usage bias than translation selection in some viruses [47,48]. This inconsistency may be because the virus genes are more extensively mutated than the plant *XET* genes. In other words, *XET* genes may be relatively conserved.

## 4. Materials and Methods

### 4.1. Sequence Data

We downloaded all the available *XET* gene sequences from the NCBI GenBank database (http://www.ncbi.nlm.nih.gov/, accessed on 1 January 2023) and analyzed their codon usage bias. The details of the available 20 available *XETs* are provided in Table 1.

### 4.2. Analysis of Codon Usage in XET Coding Sequences

We used several indicators to analyze the *XET* coding sequence codon usage. The codon adaptation index (CAI), relative synonymous codon usage (RSCU), the effective number of codons (ENc), codon bias index, frequency of optimal codon usage, hydrophobicity, aromaticity, as well as the T3s, C3s, A3s, G3s, GC, and GC3s contents were calculated using the CodonW 1.4.4 program (http://codonw.sourceforge.net/, accessed on 1 January 2023). The CAI analysis of *XET* genes was performed using the CAIcal server [49]. CAI, which is the geometric mean of the relative use of codons in genes, is used to measure the adaptability of genes to the codon usage of high-expression genes [38].

The correlation between nucleotide content was calculated using SPSS 20.0 statistical software. A Pearson correlation coefficient was calculated. ENc value was calculated to measure the degree of deviation from equal use of synonymous codons of the ORF of the *XET* genes. The ENc value, reflecting the degree of codon usage bias, ranges from 20 (when only one synonymous codon is selected for the corresponding amino acid) to 61 (when all synonymous codons are used identically) [50]. When the ENc value is greater than 35, the codon usage deviation is considered to be low [2].

### 4.3. Phylogenetic and Hierarchical Cluster Analyses

The default parameters of MEGA 7.0.12 were used to construct the phylogenetic tree based on the 20 *XET* coding sequences, and the neighbor-join (NJ) method was used with a bootstrap value of 1000 replicates [51]. Specific parameters were as follows: test of phylogeny was the bootstrap method, substitutions type was amino acid, substitution model was the Poisson model, rates among sites were uniform rates, pattern among lineages was the same (homogeneous), and gaps/missing data treatment was complete deletion. The RSCU values underwent a hierarchical cluster analysis.

### 4.4. Comparison of the XET Codon Usage Patterns

RSCU represents the ratio between the observed usage frequency of a codon in a gene sample and the expected usage frequency in the synonymous codon family, assuming that all codons of a particular amino acid are used equally. The RSCU (i.e., ratio of the observed codon usage to the expected usage [52]) was used to investigate the overall synonymous codon usage bias among the 20 *XET* genes. In a comparison of the *XET* codon usage pattern, if the RSCU value for the polyprotein-coding region of *XET* and that of the same codon were both <1.0, >1.5 or between 1.0 and 1.5, their codon usage patterns were judged to be similar. In addition, synonymous codons with RSCU values >1.5 and <1.0 are treated as high-frequency codons and low-frequency codons, respectively [43]. The preferred codons, which were defined as those whose ΔRSCU > 0.08, were analyzed as described by Zhang et al. [53]. 

The RSCU index was calculated as follows (*G_ij_* is the observed number of the *i*th codon for the *j*th amino acid, which has *N_i_* kinds of synonymous codons).
(1)$$\mathrm{RSCU}=\frac{Gij}{\sum_{j}^{Ni}Gij}Ni$$

### 4.5. Correspondence and Multidimensional Preference Analyses

The mathematical procedure of correspondence analysis converts the RSCU values into a series of dimensional factors, and the results can be used to analyze major trends in codon usage patterns among different samples. Each gene is represented by a 59-dimensional variable, and each dimension matches the RSCU value of a codon, excluding AUG, UGG, and stop codon. Correspondence analysis was performed using the CodonW 1.4.4 program (http://codonw.sourceforge.net/, accessed on 1 January 2023), and correlation analysis was performed for the first two axes (Axis 1 and Axis 2) of correspondence analysis [7]. Principal component analysis of the codon usage frequencies and multiple correspondences and multidimensional preference analyses of the RSCU value for each codon of the 20 *XET* genes were conducted using SPSS 20.0 program.

## 5. Conclusions

In conclusion, codon usage was different, and different *XET* genes had different codon usage patterns. The codon usage bias of most *XET* genes is weak, but there was also some codon usage bias. AGA, AGG, AUC, and GUG were the top four codons (RSCU > 1.5) in the 20 *XET* genes. *CitXET* had a stronger codon usage bias, and there were eight optimal codons of *CitXET* (i.e., AGA, AUU, UCU, CUU, CCA, GCU, GUU, and AAA). *PgXET*, *VlXET*, *VrXET*, *ZmXET*, *CitXET*, *BrfXET*, *BrpXET*, and *MtXET* have strong codon biases. Translational selection and base composition, followed by mutation pressure and natural selection, may be the most important factors affecting codon usage of 20 *XET* genes. Although codon usage bias is not necessarily considered in traditional phylogenetic analyses, the data presented here clarify the codon usage patterns for 20 *XET* genes. Our findings may be useful for comprehensively characterizing the factors mediating genetic evolution.

## Figures and Tables

**Figure 1 ijms-24-06108-f001:**
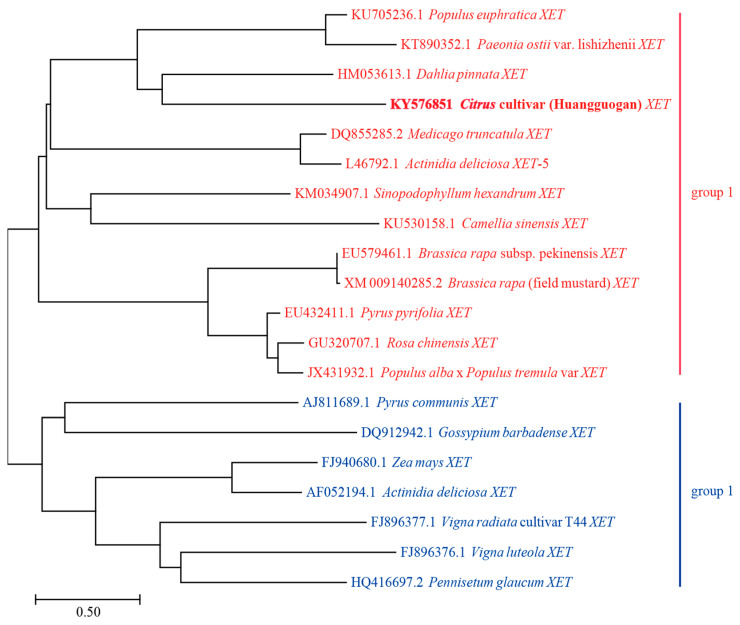
Phylogenetic tree based on the XET coding sequences. The tree was generated by the neighbor-joining (NJ) method using the MEGA 7.0.12 program. The bootstraps values were calculated with 1000 replicates.

**Figure 2 ijms-24-06108-f002:**
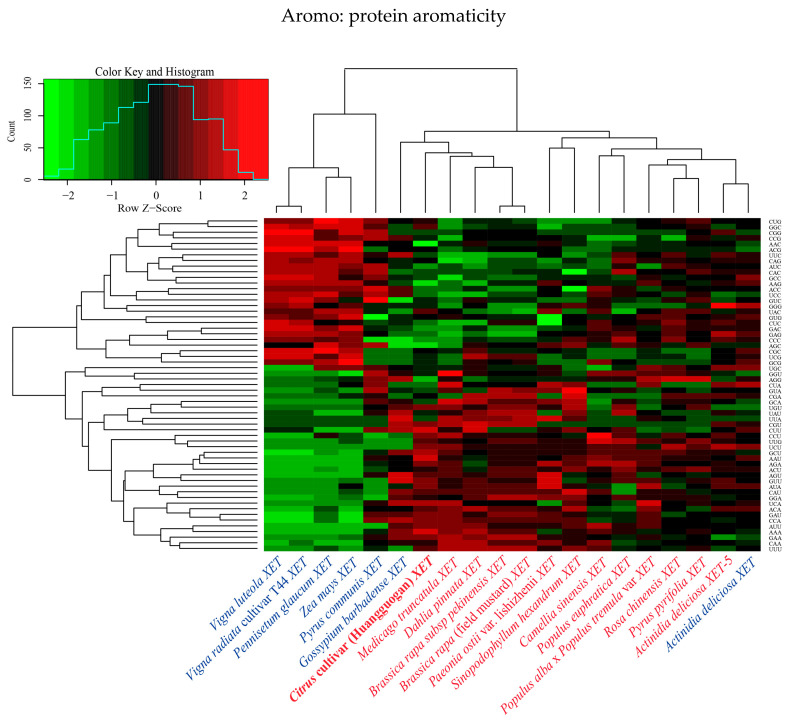
Hierarchical cluster analysis of 59 synonymous codons in 20 *XET* genes.

**Figure 3 ijms-24-06108-f003:**
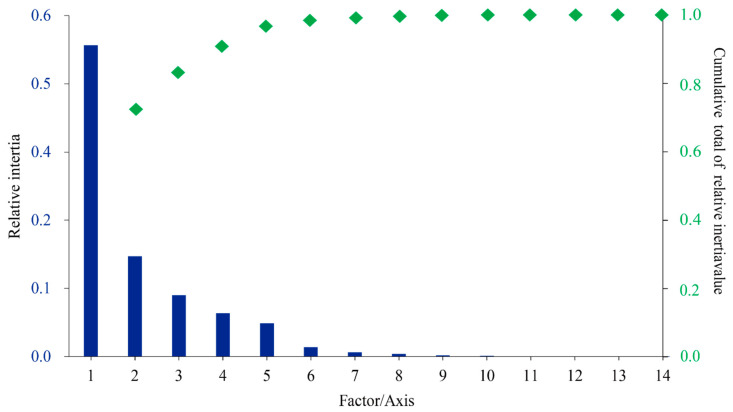
Correspondence analysis of the codon usage frequencies in 20 *XET* genes.

**Figure 4 ijms-24-06108-f004:**
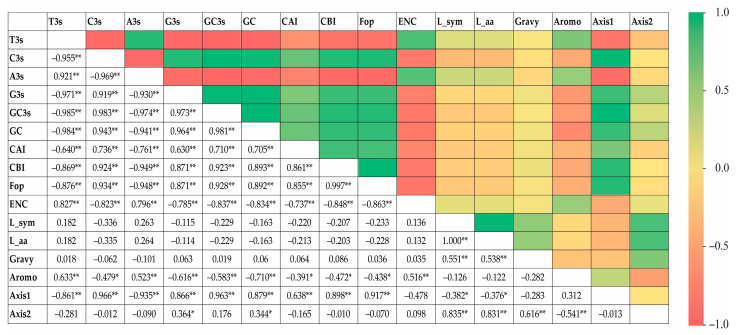
Correlations (r) between gene positions along the first two major axes of XET codon usage indices and synonymous codon usage bias. * means significant, ** means extremely significant.

**Figure 5 ijms-24-06108-f005:**
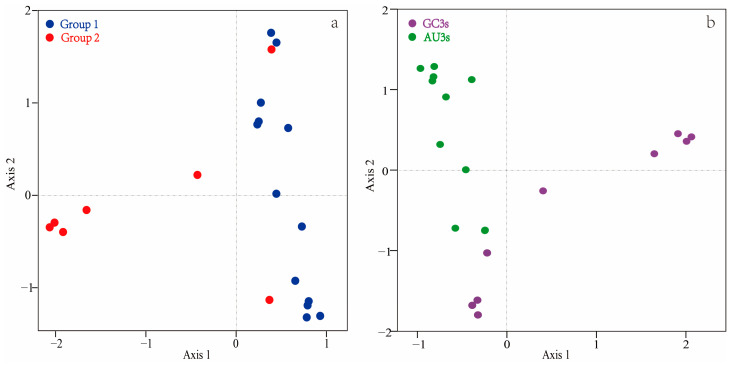
Multiple correspondence analysis of relative synonymous codon usage in the *XET* genes. Multiple correspondence analysis of Group 1 and 2 in the results of Figure 1 (**a**), multiple correspondence analysis of GC3_S_ and AU3_S_ (**b**).

**Figure 6 ijms-24-06108-f006:**
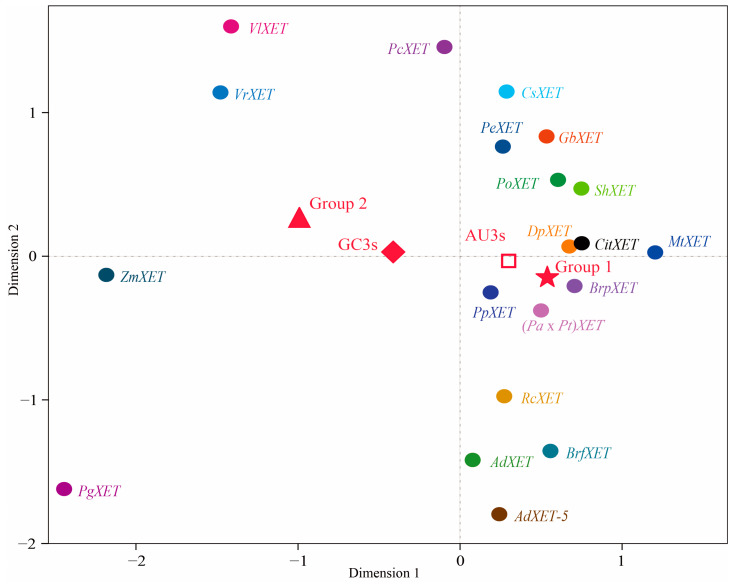
Multidimensional preference analysis of the relative synonymous codon usage for each codon of the 20 *XET* genes.

**Figure 7 ijms-24-06108-f007:**
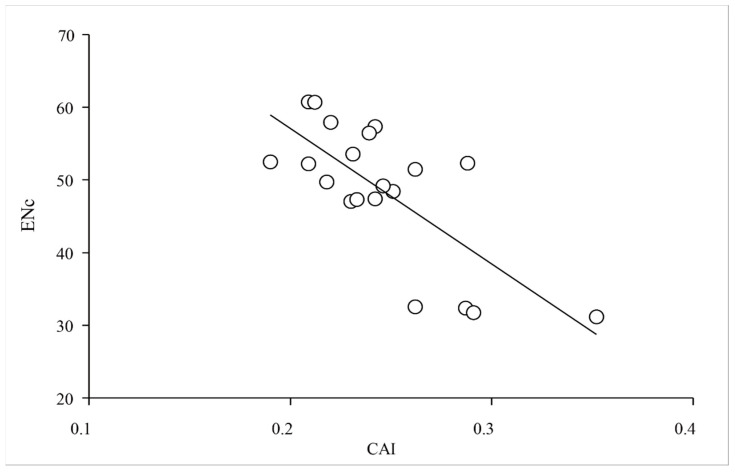
Correlation between codon adaptation index (CAI) and the effective number of codons (ENc).

**Table 1 ijms-24-06108-t001:** Accession numbers and characteristics of the analyzed *XET* coding sequences.

No.	Accession Number	Genes	U3s/%	C3s/%	A3s/%	G3s /%	GC3s/%	GC/%	CAI	CBI	Fop	ENc	Gravy	Aromo
1	KY576851	*Citrus* cultivar (Huangguogan) *XET*	43.70	27.56	36.54	22.51	37.7	41.7	0.242	0.046	0.46	47.39	−0.39	0.18
2	KU705236.1	*Populus euphratica XET*	36.36	41.96	21.24	26.21	53.4	46.2	0.251	0.140	0.52	48.40	−0.52	0.18
3	AF052194.1	*Actinidia deliciosa XET*	26.54	43.60	22.29	38.04	61.8	51.0	0.239	0.071	0.47	56.46	−0.42	0.17
4	L46792.1	*Actinidia deliciosa XET*-5	28.99	41.18	22.28	36.67	59.4	50.4	0.231	0.083	0.48	53.56	−0.50	0.17
5	XM_009140285.2	*Brassica rapa* (field mustard) *XET*	39.34	25.82	37.88	24.04	38.5	44.9	0.208	−0.058	0.39	60.70	−0.37	0.15
6	EU579461.1	*Brassica rapa* subsp. *pekinensis XET*	39.34	25.82	38.38	23.50	38.1	44.9	0.209	−0.058	0.39	60.71	−0.38	0.15
7	KU530158.1	*Camellia sinensis XET*	39.78	40.86	22.67	23.88	50.0	44.8	0.242	0.054	0.46	57.34	−0.41	0.17
8	HM053613.1	*Dahlia pinnata XET*	38.27	29.63	35.53	22.16	40.9	46.8	0.220	−0.019	0.41	57.91	−0.35	0.14
9	DQ912942.1	*Gossypium barbadense XET*	39.68	30.95	33.00	26.67	43.2	45.5	0.288	0.092	0.48	52.30	−0.57	0.16
10	DQ855285.2	*Medicago truncatula XET*	48.72	20.94	44.85	15.08	27.4	38.8	0.230	−0.067	0.39	47.04	−0.53	0.16
11	KT890352.1	*Paeonia ostii* var. lishizhenii *XET*	32.86	37.14	39.25	22.11	45.3	43.9	0.209	0.020	0.45	52.22	−0.68	0.18
12	HQ416697.2	*Pennisetum glaucum XET*	5.53	67.66	2.43	47.50	93.4	67.5	0.352	0.357	0.64	31.14	−0.44	0.14
13	JX431932.1	*Populus alba* x *Populus tremula* var *XET*	29.00	38.10	30.00	33.52	54.4	48.0	0.218	0.038	0.46	49.73	−0.45	0.17
14	AJ811689.1	*Pyrus communis XET*	22.35	49.41	22.67	36.11	65.4	54.3	0.246	0.085	0.49	49.18	−0.90	0.16
15	EU432411.1	*Pyrus pyrifolia XET*	28.14	42.42	23.20	37.36	60.1	49.7	0.262	0.148	0.51	51.44	−0.45	0.16
16	GU320707.1	*Rosa chinensis XET*	26.81	45.11	22.61	34.22	61.2	50.3	0.233	0.126	0.50	47.28	−0.53	0.16
17	KM034907.1	*Sinopodophyllum hexandrum XET*	35.11	29.79	40.88	26.00	42.0	44.4	0.190	−0.146	0.36	52.47	−0.60	0.16
18	FJ896376.1	*Vigna luteola XET*	2.38	68.25	0.00	53.85	97.9	67.8	0.291	0.309	0.61	31.75	−0.45	0.14
19	FJ896377.1	*Vigna radiata* cultivar T44 *XET*	2.40	68.00	0.93	53.92	97.2	67.1	0.287	0.301	0.60	32.36	−0.48	0.14
20	FJ940680.1	*Zea mays XET*	1.68	59.66	2.88	58.91	96.3	69.5	0.262	0.313	0.61	32.54	−0.41	0.12

U3s: frequency of the nucleotide U at the third codon position; C3s: frequency of the nucleotide C at the third codon position; A3s: frequency of the nucleotide A at the third codon position; G3s: frequency of the nucleotide G at the third codon position; GC3s: frequency of the nucleotides G + C at the third codon position; GC: G + C content; CAI: codon adaptation index; CBI: codon bias index; ENc: effective number of codons; Gravy: protein hydrophobicity.

**Table 2 ijms-24-06108-t002:** Relative synonymous codon usage values and number of codons for the 20 *XET* genes (4500 codons).

Amino Acid	Codon	Number	RSCU	Amino Acid	Codon	Number	RSCU
Phe	UUU *	105	0.58	Ala	GCU *	106	1.42
	UUC	221	1.42		GCC	104	1.40
Leu	UUA *	28	0.63		GCA *	50	0.64
	UUG *	57	1.14		GCG *	48	0.54
	CUU *	47	1.03	Tyr	UAU *	73	0.54
	CUC	68	1.42		UAC	182	1.46
	CUA	21	0.43	His	CAU *	44	0.83
	CUG	67	1.36		CAC	58	1.18
Ile	AUU *	76	0.94	Gln	CAA *	85	0.82
	AUC	89	1.55		CAG	132	1.18
	AUA *	42	0.51	Asn	AAU *	66	0.62
Val	GUU *	67	0.96		AAC	132	1.38
	GUC	60	0.94	Lys	AAA *	99	0.71
	GUA *	38	0.60		AAG	146	1.29
	GUG	113	1.50	Asp	GAU *	153	0.86
Ser	UCU *	73	1.24		GAC	183	1.14
	UCC	68	1.44	Glu	GAA *	59	0.62
	UCA	49	0.87		GAG	115	1.38
	UCG	42	0.72	Cys	UGU *	32	0.64
	AGU *	39	0.77		UGC	47	1.06
	AGC	51	0.96	Arg	CGU *	25	0.45
Pro	CCU *	56	0.85		CGC	51	0.92
	CCC	58	1.03		CGA *	22	0.39
	CCA *	76	1.19		CGG	30	0.61
	CCG	49	0.93	Arg	AGA *	86	1.89
Thr	ACU *	84	1.24		AGG *	84	1.74
	ACC	74	1.17	Gly	GGU *	77	0.97
	ACA *	69	1.02		GGC	112	1.06
	ACG	36	0.57		GGA *	99	1.15
					GGG	77	0.82

* Putatively identified as preferred codons. Underlined values indicate high-frequency codons (RSCU > 1.5).

**Table 3 ijms-24-06108-t003:** Relative synonymous codon usage values and number of codons for *CitXET* genes (300 codons).

Amino Acid	Codon	Number	RSCU	Amino Acid	Codon	Number	RSCU
Phe	UUU *	13	0.90	Ala	GCU *	10	2.67
	UUC	16	1.10		GCC	3	0.80
Leu	UUA *	0	0.00		GCA *	1	0.27
	UUG *	5	1.50		GCG *	1	0.27
	CUU *	8	2.40	Tyr	UAU *	3	0.35
	CUC	2	0.60		UAC	14	1.65
	CUA	0	0.00	His	CAU *	4	1.33
	CUG	5	1.50		CAC	2	0.67
Ile	AUU *	9	1.59	Gln	CAA *	6	0.86
	AUC	3	0.53		CAG	8	1.14
	AUA *	5	0.88	Asn	AAU *	8	1.33
Val	GUU *	7	1.65		AAC	4	0.67
	GUC	3	0.71	Lys	AAA *	17	1.62
	GUA *	1	0.24		AAG	4	0.38
	GUG	6	1.41	Asp	GAU *	16	1.45
Ser	UCU *	7	2.10		GAC	6	0.55
	UCC	2	0.60	Glu	GAA *	6	1.09
	UCA	6	1.80		GAG	5	0.91
	UCG	1	0.30	Cys	UGU *	4	1.30
	AGU *	4	1.20		UGC	2	0.70
	AGC	0	0.00	Arg	CGU *	3	0.90
Pro	CCU *	5	1.33		CGC	2	0.60
	CCC	1	0.27		CGA *	1	0.30
	CCA *	7	1.87		CGG	0	0.00
	CCG	2	0.53	Arg	AGA *	13	3.90
Thr	ACU *	6	1.50		AGG *	1	0.30
	ACC	2	0.50	Gly	GGU *	4	0.70
	ACA *	6	1.50		GGC	8	1.50
	ACG	2	0.50		GGA *	7	1.30
					GGG	3	0.60

* Putatively identified as preferred codons. Underlined values indicate high-frequency codons (RSCU > 1.5). Fraction: proportion of all codons represented by a specific codon for a particular amino acid.

## Data Availability

Not applicable.

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
