# Peer review of "Analysis of Codon Usage Bias in Xyloglucan Endotransglycosylase (XET) Genes"

_ijms, 2023, doi:10.3390/ijms24076108_

Round 1

Reviewer 1 Report

The Methods section must be rewritten with more details for reproducibility.

The discussion section must be in deep and interpret all results obtained 

Reviewer 2 Report

To,

The Editor,

IJMS, MDPI,

Manuscript ID: ijms- 2156927

Subject: Submission of comments of the manuscript in “IJMS"

Dear Editor IJMS, MDPI,

Thank you very much for the invitation to consider a potential reviewer for the manuscript (ID: ijms- 2156927). My comments responses are furnished below as per each reviewer’s comments. 

The manuscript by Xiong et al. analyzed the codon usage bias using 4,500 codons from 20 XET genes to elucidate the genetic and evolutionary patterns. Phylogenetic and hierarchical cluster analyses of coding sequences and relative synonymous codon usage (RSCU) values, respectively, revealed the 20 XET genes belonged to two groups. CitXET and DpXET were clustered in a Group 1 sub-group, and exhibited similar codon usage biases. The top four codons in the 20 XET genes were AGA, AGG, AUC, and GUG. Additionally, we identified eight optimal codons of CitXET (i.e., AGA, AUU, UCU, CUU, CCA, GCU, GUU, and AAA). Base composition (especially A and U at the third codon position) was observed to influence codon usage bias, possibly because of the specific XET gene functions. The two main factors affecting codon usage bias (i.e., Axes 1 and 2) represented 54.8% and 17.6% of the total variation, respectively. PgXET, ZmXET, VlXET, VrXET, and PcXET were biased toward codons ending with G/C. In contrast CitXET, DpXET, and BrpXET were strongly biased toward codons ending with A/U. Different XET genes generally exhibited diverse codon usage patterns, although there were also some codon usage biases. The closer the genetic distance, the more similar their codon usage preference. Our results may be useful for clarifying the codon usage bias of XET genes, and the relevant evolutionary characteristics. The analysis is comprehensive, and the data provided in this manuscript is a useful resource in the field, therefore this work is interesting and important in the field. I have several suggestions to improve this manuscript:

  1. The topic is interesting and well within the aims of the Journal, but it needs major revisions before it ca be published. In general, a careful revision of the language must be carried out as well as of the punctuation. Moreover, several paragraphs are disconnected from each other and sometimes repetitive. Here some tips to improve the work, but as already said all the work needs a more accurate revision.
  2. Abstract do not reflect the essence of the study. Abstract needs to be improved with detailed results.
  3. Introduction looks shallow, write more detail about studied issue. Introduction section do not highlight the literature and study question
  4. Overall, the amount of work is very less and the data are short and just does not meet the criteria of publication in current form (need more figures and supplementary data).
  5. The figures quality is not up to the standards. The authors are strongly recommended to prepare the good quality figures including the 1, 2 and 4.

6.    The discussion needs to be re written, in fact it looks more like an introduction. There are many things to discuss, however there is only a short description of the results at the end. 

  1. Authors must add the conclusion
  2. Line 260: Helianthus annuus  must be italic.
  3. Line 268: Ophioglossum vulgatum L. must be italic.
  4. Line 296-297: Cuscuta reflexa must be italic.
  5. Line 313: Zea mays must be italic.
  6. Line 359: Silene latifolia must be italic.
  7. Line 359: Molecular biology and evolution replaced with Mol. Biol. Evol.
  8. Line 369: Triticum aestivum L. must be italic.
  9. Line 369: Journal of Integrative Plant Biology replaced with J. Integr. Plant Biol.
  10. Overall, the MS is poorly organized and cannot be recommended to publish this MS.

Reviewer 3 Report

In the present work, authors have made a detailed analysis of codon usage bias in a xyloglucan endotransglycosylase (XET) genes. In general, the manuscript is well-presented and the analysis done properly. The authors argue that their findings may be useful for comprehensively characterizing the factors mediating genetic evolution. In my opinion they should improve this discussion in the manuscript, explaining this main contribution. Mainly, XET genes didn’t present codon usage bias and one CitXET presents a strong bias, what is the hypothesis to explain it. In order, what is the importance of this kind of result concerning XET genes.

Material and methods

Please add an explanation of the criteria used to choose XET genes. Is the number of genes, 20 genes, enough for this kind of analysis?

Reviewer 4 Report

Comments to the Authors for ijms-2156927

The MS “ijms-2156927” entitled “Analysis of codon usage bias in xyloglucan endotransglycosylase (XET) genes” by Bo Xiong et al., presents a bioinformatic analysis regarding the codon usage patterns of 20 XET genes based on the cloned CitXET gene from the Citrus cultivar Huangguogan. Correspondence analysis used is an ordination technique that identifies the major trends in the variation of the data and distributes genes along continuous axes in accordance with these trends. CA has the advantage that it does not assume that the data falls into discrete clusters and can therefore represent continuous variation accurately.

The core data presented are quite interesting, since it lay out the foundation to understand the codon usage bias of XET genes, which may be useful to characterize the factors mediating genetic evolution. Thus, it is a welcome manuscript and I believe that it is worthy of publishing in the “IJMS”. However, the MS in its current form is premature. While the objectives of the research work are clear and the experimental approach appropriate, several points need clarification and/or better presentation to make the article more robust, reader-friendly, and suitable for publication.

Major points:

1) In “Materials and Methods” as well as in “Results”, more information and a better presentation are needed regarding the phylogenetic tree. What parameters were used for constructing the tree? Is this a rooted or unrooted tree? Does it represent a consensus tree? Was bootstrap analysis applied? How many replicates?  The tree should contain the bootstrap analysis numbers (bootstrap support values from at least 1.000 replicates). In other words, a more comprehensive bootstrap phylogenetic analysis should be presented.

2) The “Discussion” is a short repetition of the data presented in “Results. The “Discussion” should not simply repeat what has already been said in “Results” but put the research data into context with a refined merge of the current findings with previous data in this scientific field. Must be significantly elaborated.

Minor language issues:

Line 34: …of the errors also relies on the frequency of diferent different codons[2].

Line 38: …but the codon usage bias was strongestrong[6].

Line 49: …which belong to the glycosyl hydrolase family 16[14],…

Line 174: …and PcXET and GC3s suggests that the codon usage…

Line 218: Rephrase and correct the sentence.  … In conclusion, Similar conclusions were reached…

I hope that the Authors find these comments useful to improve their manuscript.

Round 2

Reviewer 2 Report

Dear Editor,

Thank you for providing the opportunity to review the revised manuscript. The manuscript is improved considerably after revision according to the reviewer's comment. Now this study is a suitable contribution to the IJMS. I recommend the manuscript for publication.

Thank you

With best regards